# Empirical Evidence of Reduced Wildfire Ignition Risk in the Presence of Strong Winds

**Assaf Shmuel *** and **Eyal Heifetz**

Department of Geophysics, Porter School of the Environment and Earth Sciences, Tel Aviv University, Tel Aviv 69978, Israel; eyalh@tauex.tau.ac.il
* Correspondence: assafshmuel91@gmail.com

**Abstract:** Anyone who has tried lighting a campfire on a windy day can appreciate how difficult it could be. However, despite real-life experience and despite laboratory experiments which have demonstrated that fire ignition risk dramatically decreases beyond a certain wind threshold, current fire weather indices (FWIs) do not take this effect into account and assume a monotonic relation between wind velocity and ignition risk. In this paper, we perform a global analysis which empirically quantifies the probability of ignition as a function of wind velocity. Using both traditional methods (a logistic regression and a generalized additive model) and machine learning techniques, we find that beyond a threshold of approximately 3–4 m/s, the ignition risk substantially decreases. The effect holds when accounting for additional factors such as temperature and relative humidity. We recommend updating FWIs to account for this issue.

**Keywords:** machine learning; fire weather indices; wind velocity; forest management

## 1. Introduction

Wind velocity is a central determinant of wildfire propagation rate. Numerous studies have demonstrated the effect of strong winds on wildfire expansion rate. For example, Ref. [1] study the development of wildfires across the US and demonstrate how high winds drive rapid fire growth and overcome many restraining factors: strong winds limit aerial fire suppression efforts and completely ground aircrafts at wind over 15 m/s [2]; the strong winds increase flame lengths, encouraging the spread of the fire and making it difficult for fire suppression crews to perform direct attack [3]; finally, strong winds lead to increased spotting and support convective pre-heating, which are factors that have been identified as leading to firestorms and fatalities [4]. Many additional studies recognize wind velocity as one of the major factors that leads to extreme fires and rapid wildfire growth (e.g., [5]). A recent study has found that wind velocity is the most important factor in determining the wildfire propagation rate, making it possible to estimate the wildfire propagation rate as roughly 10% of the 10 m open wind velocity [6].

The positive correlation of strong winds and rapid wildfire propagation is not only acknowledged in scientific research but is also reflected in all of the common fire weather indices (FWIs). In the National Fire Danger Ratings System (NFDRS), the most common FWI in the US [7], wind speed only directly affects the spread component (SC), which is numerically equal to the theoretical ideal rate of spread expressed in feet per minute [7]. The index monotonically increases with wind speed [8]. The McArthur Forest Fire Danger Index [9] includes a simple exponent which doubles the FWI when the wind velocity increases at about 20 km/h [10,11]. In the Canadian Fire Weather Index [12], the influence of wind over the Initial Spread Index (ISI) is described using a similar exponent. Ref. [12] notes that this exponent was derived empirically, and its validity at high wind speed may be uncertain [12].

While strong winds undoubtedly lead to a faster expansion rate in existing wildfires, we claim that the probability of wildfire ignition is substantially reduced in the presence of

strong winds. Intermediate wind of approximately 3 m/s increases oxygen supply and has been shown to encourage wildfire ignition more than weak winds of 0–1 m/s (e.g., [13–18]). Beyond a certain wind velocity threshold, effects which decrease the probability of ignition may apply. For example, studies have found that beyond a wind velocity of 4–6 m/s, lit cigarettes are most likely to be carried by the wind [19], and beyond a threshold of 4 m/s, they are likely to be extinguished [20]. Strong winds could cool firebrands and extinguish ignition sources [21]. Although the wind increases oxygen supply, its cooling effect can reduce the transition from smoldering to flaming beyond a certain threshold [22].

The effect of strong winds on ignition has been demonstrated in several laboratory experiments. Ref. [23] performed 2500 ignition experiments to estimate the probability that cigarette butts would ignite a fuel bed in the presence of different weather conditions. They found that up to a certain threshold, stronger wind increases the probability of ignition, but beyond that threshold, higher wind velocities reduce ignitions. An additional laboratory experiment in cotton balls ignition also demonstrated a negative correlation between wind and ignition probability [24]. Ref. [25] developed a physical model which forecasts a non-monotonic correlation between wind velocity and ignition risk.

While the lower probability of human-caused wildfires ignition can be explained by the aforementioned factors, the ignition mechanism of lightning-caused wildfires is entirely different. Scholars have studied the necessary meteorological conditions for lightning-caused wildfire ignition in several regions. Although some meteorological conditions are different in such circumstances (e.g., higher RH values), the probability of lightning-caused wildfire ignition is highest at intermediate range wind velocity values, which is similar to human-caused wildfires. For example, Ref. [26] find that lightning-caused wildfires in central Brazil are typically ignited at a wind speed of approximately 2 m/s.

In this study, we aim to empirically examine the effect of strong winds on ignition probability using a global dataset. The paper is organized as follows. We begin by providing a detailed summary of our data, which is followed by a description of the methods we apply in the research. We then present an analysis of the probability of wildfire ignition as a function of wind velocity, either based on traditional statistical models or on machine learning (ML) techniques. In the final section, we discuss the contribution, implications and limitations of the study, and propose directions for future research.

## 2. Research Design

### 2.1. Data

Our target variable is the daily burned area at a 0.25° resolution global land grid. We produce this grid based on the dataset published in [27], which includes global wildfire data in a daily resolution. We use the entire observations of the year 2016, which include a total of 1,024,926 different wildfires. We assign the value of 1 to regions in which a wildfire ignited on the day of the observation and 0 otherwise. Approximately 1.4% of the observations were assigned the value of 1.

The independent variables in our models include meteorological factors, anthropogenic factors, and fuel loads. The meteorological data are taken from the ERA5 hourly reanalysis dataset [28]. We use 2-m temperature, relative humidity and 10 m wind velocity.

Previous studies have shown that population density has a substantial effect on wildfire occurrence (e.g., [29]). We include population density based on the dataset of the Center for International Earth Science Information Network [30]. While the original dataset is provided with a resolution of ~1 km, we calculate the mean population density in each 0.25° region.

Leaf area index (LAI) is a variable that describes the leaf material in a given location. LAI is a dimensionless variable that varies between 0 and approximately 10. LAI data at a 1/112° (~1 km) resolution are taken from [31]. The LAI data are originally separated to low and high vegetation; we include one variable describing their sum.

For plotting partial dependencies (as explained in the following section), we use two common wildfire indices: the Canadian Forest Service Fire Weather Index Rating System

(FWI) and the Australian Forest Fire Danger Index (FFDI). All data are available in 0.25° resolution and were obtained from the Copernicus Climate Change Service [32]. Table 1 summarizes the features and data sources used in the paper.

**Table 1.** Summary of features and data sources.

| Variable | Abbreviation | Source |
|---|---|---|
| daily burned classification | burned | [27] |
| 2 m temperature | temp | |
| relative humidity | RH | [28] |
| 10 m wind speed | wind_speed | |
| population density | population | [30] |
| leaf area index | LAI | [31] |
| daily fire weather index | FWI | [32] |
| daily fire danger index | FFDI | |

*2.2. Methodology*

We perform several analyses to evaluate the effect of strong winds on wildfire ignition risk. All of the analyses are performed using Python. We begin by applying traditional statistical models and fit a logistic regression [33,34] whose dependent variable is the probability of ignition. A logistic regression is a widely used statistical model when predicting binary variables (such as wildfire occurrence). A logistic regression was applied based on the following equation:

$$P_i = \frac{1}{1 + e^{-z_i}}$$

where $P_i$ is the probability of wildfire occurrence, $z_i$ is a linear function of the independent variables we use as predictors ($x_1 \ldots x\_n$):

$$z_i = \alpha + \beta_1 x_{1i} + \beta_2 x_{2i} \ldots . \beta_n x_{ni}$$

$\alpha$ is a constant and $\beta_1 \ldots \beta_n$ are the regression coefficients. We evaluate the logistic regression using Python's Scikit-learn library using its default parameters: L2 regularization and the LBGFS optimizer. In the basic model, we include five terms: wind velocity, RH, temperature, population, and LAI. The dependent variable is the occurrence of a wildfire in the spatiotemporal observation. We tested whether removing one of the independent variables could improve (decrease) the Akaike's Information Criterion (AIC) score [35] and found that it did not. To capture the expected nonlinear relation between wind velocity and ignition risk, we present a model which includes both wind velocity and its square value in addition to a model which only includes the wind velocity (increasing the number of terms to 6). We use several control variables in these analyses: relative humidity, temperature, population density and leaf area index. We emphasize that we do not include fire weather indices in this analysis, as this may lead to collinearity.

An additional method to capture the nonlinear effect of wind velocity on wildfire occurrence is the Generalized Additive Model (GAM) [36,37]. The GAM model has the form $\eta(x) = \alpha + \sigma \, f_j \, (x_j)$, where $\eta$ is either the regression function in a multiple regression or the logistic transformation of the posterior probability in a logistic regression [36]. It is recommended to use a GAM model when a model contains nonlinear effects, as GAM provides regularized and interpretable solutions in such cases. As our hypothesis is of a nonlinear relationship between wind velocity and wildfire occurrence, we develop a GAM model using the Python pyGAM library [38] and present the partial dependence plot (PDP) based on this model.

Next, we apply a machine learning model to capture more complex relations between wind velocity and wildfire ignition risk. We develop a model based on an Extreme Gradient

Boosting (XGBoost) classification model [39]. XGBoost is an advanced implementation of gradient boosting algorithm—an ensemble tree method that uses gradient descent to boost weak learners [40,41]. The XGBoost implementation has been widely used in numerous ML applications and proven its outstanding performance (e.g., [42–44]). We use the following hyperparameters: 100 estimators, maximum tree depth of 6, learning rate of 0.1, gbtree booster, and a 'binary:logistic' objective. As the number of features is relatively small, hyperparameter tuning had little effect on the models. The hyperparameter tuning included changing the number of estimators in the range of 50–200, the maximum tree depth in the range of 4–8, and the learning rate in the range of 0.05–0.2. We do not present the full results in the paper, as these hyperparameters had little effect on the model.

One of our primary concerns is that the hypothesized negative correlative between strong wind and ignition probability is not a direct effect of the wind but rather that a third variable is involved. For example, wildfires are significantly less likely to occur during winter days, but winter days are commonly characterized by strong winds. We perform several different analyses to address this concern. First, in addition to the PDP over the entire data, we perform three similar analyses over subsets of the data in which the RH is relatively low (<20, <30, and <40) and three more analyses for subsets in which the fire weather index is relatively high (>25, >50 and >75). By using these subsets of the data, we remove days in which wildfires are less likely to ignite, such as winter days. We acknowledge that FWI values are affected by wind velocity, which may somewhat affect our conclusions; however, we have several justifications for this analysis: (a) we also present an analysis in which the subsets of the data are determined by RH, which is not directly affected by wind velocity; (b) the correlation between FWI and wind velocity was relatively low (0.2 for FWI–wind velocity compared to 0.41 for FWI–temperature and $-0.78$ for FWI–RH).

In addition, we present several three-dimensional PDPs which present the dependence of ignition on wind velocity as well as FWI, FFDI or RH. These analyses demonstrate the influence of wind velocity under different fire danger levels. The PDPs are performed using Python's Scikit package [45].

## 3. Results

### 3.1. Probability of Ignition Based on Statistical Models

Table 2 presents the probability of ignition based on a logistic regression as a function of wind velocity, squared wind velocity and several control variables (RH, temperature, population, and LAI). When assuming a linear dependence on wind velocity (Model #1), we find that an increase of 1 m/s in wind velocity reduces the probability of ignition by 14%. By adding a variable representing the squared value of wind velocity, the wind velocity becomes positively correlated with fire ignition probability, while the squared wind velocity variable is negative. This result verifies the expected dependence of ignition probability on wind velocity—at very low wind speeds, an increase in wind velocity increases the probability of ignition by approximately 8% per 1 m/s increase in wind velocity. Beyond a certain threshold, however, wind velocity reduces the probability of ignition. For example, an increase from 4 to 5 m/s reduces the probability of ignition by approximately 28%.

We next present the results of the GAM model. Figure 1 presents a PDP of wind velocity based on the GAM model. Winds beyond 13 m/s are rare in the data, as is evident from the wide confidence interval. We therefore only address the range between 0 and 13 m/s. These results support the hypothesis of reduced wildfire ignition in the presence of strong winds. In fact, this decrease is much more substantial than the known increase in wildfire ignition risk at low wind velocities (below 3 m/s), which is hardly statistically significant in the current analysis.

**Table 2.** Coefficients and odds ratios for the probability of ignition based on a logistic regression.

| Model | #1 | | #2 | |
|---|---|---|---|---|
| | **Coefficient** | **Odds Ratio** | **Coefficient** | **Odds Ratio** |
| **Wind Velocity** | −0.15 *** | −0.14 *** | 0.07 *** | 0.08 *** |
| | (0.00) | (0.00) | (0.00) | (0.00) |
| **Wind Velocity Squared** | - | - | −0.04 *** | −0.04 *** |
| | | | (0.00) | (0.00) |
| RH | −0.05 *** | −0.05 *** | −0.05 *** | −0.05 *** |
| | (0.00) | (0.00) | (0.00) | (0.00) |
| Temperature | −0.007 *** | −0.007 *** | −0.008 *** | −0.007 *** |
| | (0.00) | (0.00) | (0.00) | (0.00) |
| Population | −0.0005 *** | −0.0005 *** | −0.0004 *** | −0.0004 *** |
| | (0.00) | (0.00) | (0.00) | (0.00) |
| LAI | 0.22 *** | 0.25 *** | 0.22 *** | 0.25 *** |
| | (0.00) | (0.00) | (0.00) | (0.00) |

Coefficients and odds ratios for the probability of ignition. The odds ratios are calculated as the exponentiated coefficients of a logistic regression subtracted by 1; $p$-values in parentheses. *** $p < 0.01$.

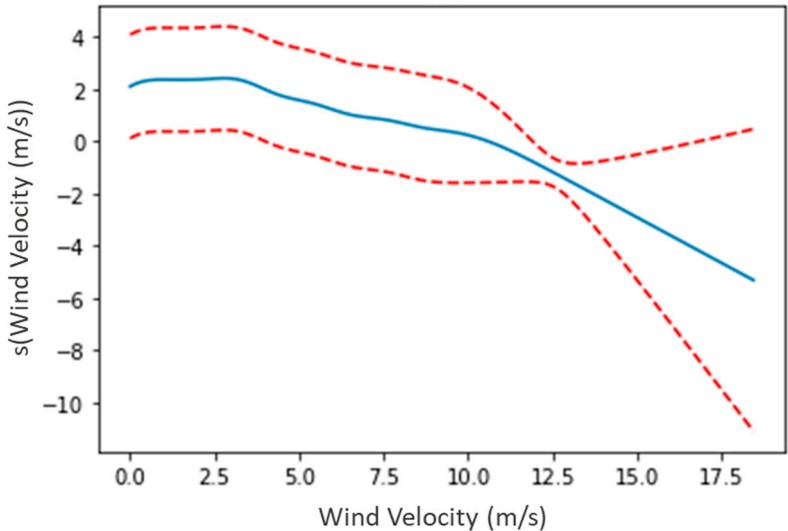

**Figure 1.** PDP for wind velocity based on the GAM model. A PDP analysis of the effect of wind velocity on the probability of wildfire ignition based on a GAM model. The red line presents the [25%, 75%] confidence interval.

### 3.2. Probability of Ignition Based on a Machine Learning Model

We now present a wind velocity PDP for the probability of ignition based on a machine learning model. The results are presented in Figure 2. The black line describes the partial dependence when accounting for the full dataset. As expected, the probability of ignition increases up to a wind velocity of 3 m/s and decreases for stronger winds. The three remaining lines describe a similar analysis when accounting for subsets of the data which only include low RH values or high FWI values. While the partial dependence values are not identical, the shape of the dependence plot is similar: for all four analyses, we find an initial increase in ignition probability up to a certain threshold beyond which we receive a strong decrease in ignition probability. In addition, in all four analyses, the threshold is set at approximately 3 m/s.

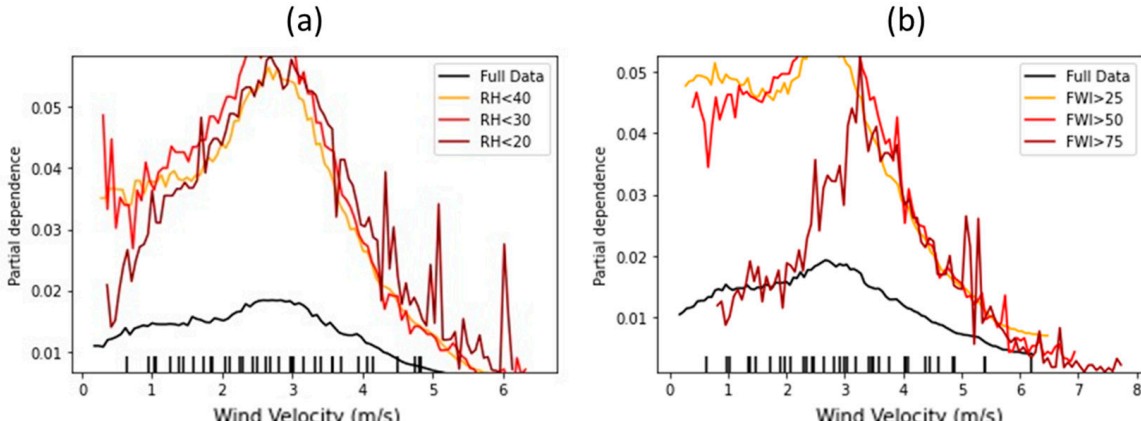

**Figure 2.** Probability of wildfire ignition as a function of wind velocity—different subsets of the data. A PDP analysis of the effect of wind velocity on the probability of wildfire ignition. The figure is based on an XGBoost model with 100 estimators. (**a**) we divide the data by RH; (**b**), we divide the data by FWI values. A similar effect is obtained for all subsets of the data.

We now present a similar analysis using a contour plot to clearly demonstrate the dependence of the target variable on wind velocity and fire danger. Figures 3–5 present the probability of ignition based on wind velocity and one of three control variables: FWI, FFDI and RH. Similar results are obtained in all three analyses: the probability of ignition decreases beyond the threshold of 3–4 m/s.

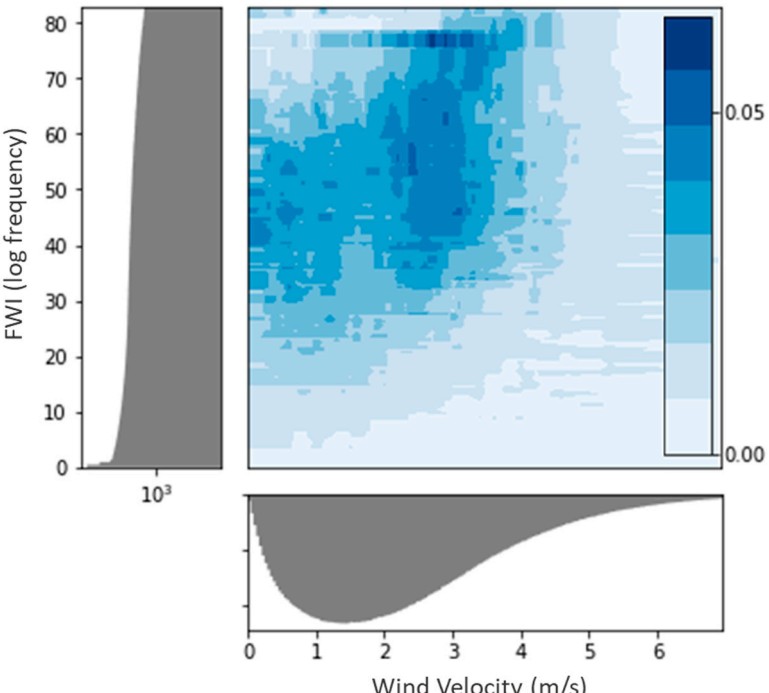

**Figure 3.** Probability of wildfire ignition as a function of wind velocity and FWI. Estimation of ignition probability based on wind velocity and FWI values. The histograms on the bottom and left sides of the figure present the distributions of wind velocity and FWI values. The figure is based on an XGBoost model with 100 estimators. As expected, the probability increases with FWI for all wind velocity values. For substantial FWI values (FWI > 30), the probability of wildfire ignition substantially decreases above a threshold of approximately 3–4 m/s.

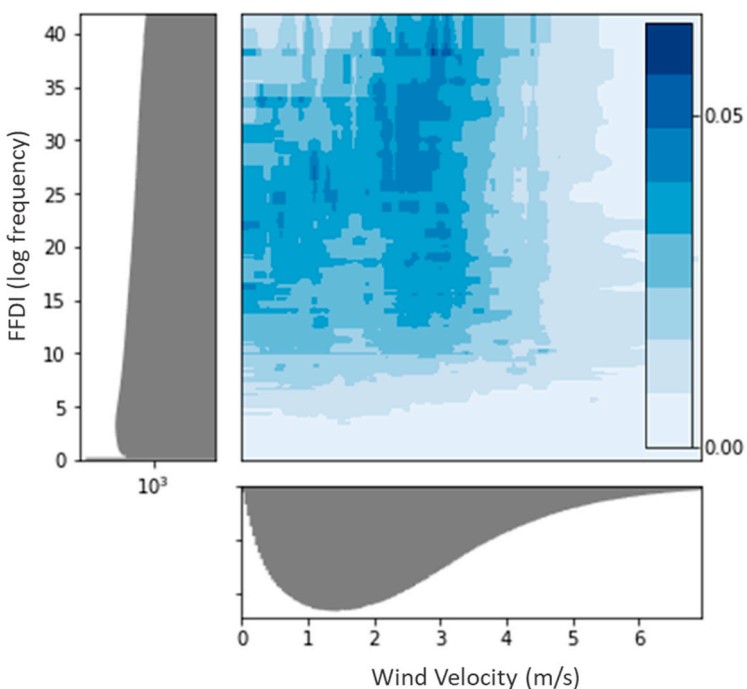

**Figure 4.** Probability of wildfire ignition as a function of wind velocity and FFDI. Estimation of ignition probability based on wind velocity and FFDI values. The histograms on the bottom and left sides of the figure present the distributions of wind velocity and FFDI values. The figure is based on an XGBoost model with 100 estimators. As expected, the probability increases with FFDI values for all wind velocity values. For substantial FFDI values (FFDI > 20), the probability of wildfire ignition substantially decreases above a threshold of approximately 3–4 m/s.

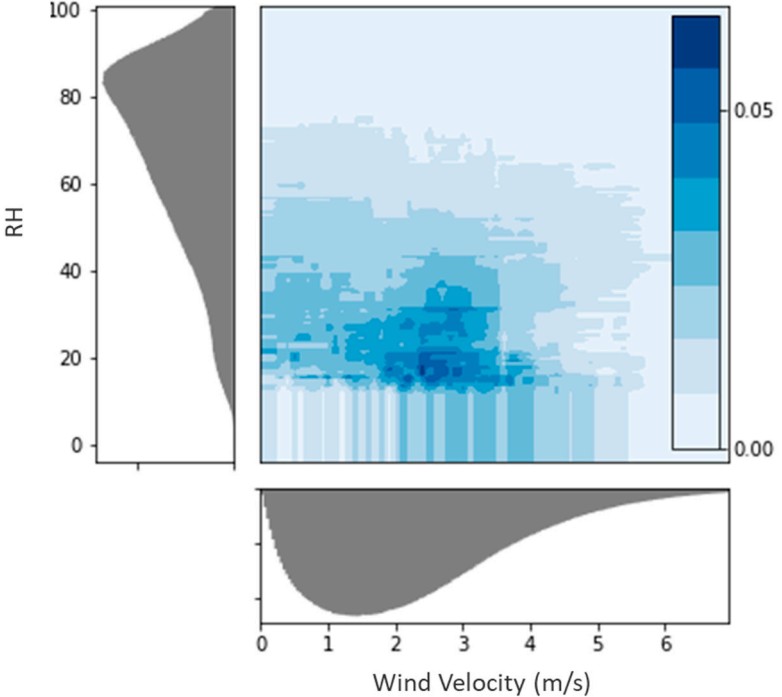

**Figure 5.** Probability of wildfire ignition as a function of wind velocity and RH. Estimation of ignition probability based on wind velocity and RH values. The histograms on the bottom and left sides of the figure present the distributions of wind velocity and RH values. The figure is based on an XGBoost model with 100 estimators. As expected, the probability is highest for low RH values. Similarly to the previous analyses, the probability of ignition decreases above the 3–4 m/s threshold.

## 4. Discussion

In this paper, we performed an empirical examination of the effect of wind velocity on the risk of wildfire ignition. We applied both logistic regressions and ML models to validate the negative effect of strong winds on wildfire ignition risk and quantify the wind velocity threshold beyond which wildfires are less likely to ignite. The results of the study show that strong winds are indeed negatively correlated to wildfire ignition risk. This result holds when accounting for additional factors which determine wildfire risk. Beyond a threshold of approximately 3–4 m/s, the probability of ignition substantially decreases and becomes almost negligible at wind velocities of around 6 m/s. These values are in line with previous laboratory studies which have demonstrated that ignition probability is reduced at similar values [24,25] and cigarette butts are normally extinguished by such winds [19,20].

Several limitations of this study should be noted. First, we performed this analysis in a 25 km resolution, possibly missing mesoscale phenomena which affect the variability of wind velocity within each region. For example, topography or canopy cover affect the distribution of wind velocity within each region. An additional limitation is that this study was performed using a daily temporal resolution; spatiotemporal observations in which the winds only occur in certain hours of the day could still be at relatively high wildfire ignition risk during some hours of the day. Future studies could examine the effect of wind in finer resolution, such as the maximal daily wind gusts.

The results of this study could have important implications for fire weather indices. To the best of our knowledge, the most common indices assume a monotonic correlation between wind velocity and fire risk—both for ignition probability and for wildfire propagation. While this assumption is appropriate for wildfire propagation, we believe that indices describing wildfire ignition risk should reflect a negative correlation with wind velocity beyond a certain wind velocity threshold. In this paper, we have demonstrated that this effect can be substantial even when controlling for additional factors. Since wind is one of the most important factors in wildfire prediction, we propose to re-examine its role in the various fire weather indices to better reflect its complex and contradictory effect between wildfire ignition and wildfire propagation.

## 5. Conclusions

We examine the effect of wind velocity on wildfire ignition risk using a global dataset. We find that strong winds decrease the probability of wildfire ignition. This result is in line previous studies which have demonstrated this effect in laboratory experiments. The negative effect of strong winds on wildfire ignition is in contrast to the (strong) positive effect of strong winds on the rate of spread of wildfires which are already burning, demonstrating the complex relationship between wind and wildfires.

**Author Contributions:** Conceptualization, A.S. and E.H.; Methodology, A.S. and E.H.; Software, A.S.; Writing—original draft, A.S.; Writing—review and editing, E.H.; Supervision, E.H. All authors have read and agreed to the published version of the manuscript.

**Funding:** This study did not receive any specific funding.

**Institutional Review Board Statement:** Not applicable.

**Informed Consent Statement:** Not applicable.

**Data Availability Statement:** No new data were created in this study. Data sharing is not applicable to this article.

**Acknowledgments:** We thank Amir Dalal, the three anonymous reviewers and the editorial team for their careful reading of our manuscript and their many insightful comments and suggestions.

**Conflicts of Interest:** The authors declare no conflict of interest.

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
