# Peer review of "Empirical Evidence of Reduced Wildfire Ignition Risk in the Presence of Strong Winds"

_fire, doi:10.3390/fire6090338_

Round 1

Reviewer 1 Report

Dear authors, the manuscript is very interesting. I believe that others initiative similar research in specific areas.

I suggest that the title ''Discussion'' should be moved and places next to ''Results'' and that ''Conclusion'' should be written in that place.

I congratulate you on your efforts and wish you much success in the future.

Best regards

Reviewer 2 Report

General comments

The introduction is very well written. I find that further work is necessary in the methods and results to ensure that the findings are real and are not artefacts of the methods.  Additional detail in the methods is also needed, as there is insufficient information to replicated the work.

The literature review for this paper is very clear and well written.  I am happy with its current form.

I find that additional information is required in the methods and results.  As written, the study could not be simply replicated.  In particular

-        There is no information on the fitting process for the statistical logistical regression, including complexity of models, model selection, number of terms included and testing.

-        The models are not presented, only the odds ratios and p values.

-        An AIC approach for variable selection would have been more appropriate.

-        It is not clear what is meant be ‘control variables’

-        The combination of wind and win^2 is not an ideal way to portray the shape of the wind curve.  If the aim is to demonstrate that the shape of the curve is more complex, a linear model is not appropriate.  Ideally a GAM would show the pattern as it exists in the data (generalized addivive model – see Hastie)

-        It is not clear what software was used.

-        Given that use of two fire weather indices, covariation should be considered.  If covariation is high, neither model will be able to isolate each indices’ contribution.

In relation to the ML model

-        The fitting parametrisation is not presented, I would not be able to repeat this analysis.  There is no indication of partitioning, tree depth etc.

-        It in not clear what is meant by ‘several models’

-        In terms of the result, the Fire weather indices include wind – there is a chance that they are being selected in preference to wind speed but provide the same information on the bad fire days.

As written, I cannot be sure that the main finding (ignition probability decreases with wind speed) is clear from the analysis.  The key issues are  1) that the statistical approach assumes curve shapes and no other forms are considered, and 2) the inclusion of indices that already encompass wind speed in the ML model mean that there could be substitution leading to a spurious result (e.g. FFDI is selected to represent high wind speed).

Reviewer 3 Report

This study empirically examined the impact of wind velocity on wildfire ignition risk and utilized both logistic regression and machine learning models to validate the adverse effect of strong winds on wildfire ignition risk. The research methods and data analysis were clear, and the results presented the intricate relationship between wind velocity and ignition probability, potentially having significant implications for fire weather indices. Overall, this is a valuable study that contributes to understanding wildfire ignition mechanisms and fire risk assessment.

I find the empirical analysis and results of this study to be reliable, but there are some potential areas for improvement and further investigation:

1. It is suggested that the authors provide a brief introduction to the machine learning analysis method used and share its applicability.

2. It is recommended to present the mathematical expressions of the regression models established and explain their meanings.

3. In the machine learning model, were the parameters adjusted? If so, it is advised to provide a detailed explanation of the parameter tuning methods and results to ensure the reliability and consistency of the model.

This study provides valuable insights for forest wildfire risk assessment, but there is room for improvement. I recommend accepting the paper with minor revisions.
